# The Prognostic Value of Eosinophil Recovery in COVID-19: A Multicentre, Retrospective Cohort Study on Patients Hospitalised in Spanish Hospitals

**DOI:** 10.3390/jcm10020305

**Published:** 2021-01-15

**Authors:** María Mateos González, Elena Sierra Gonzalo, Irene Casado Lopez, Francisco Arnalich Fernández, José Luis Beato Pérez, Daniel Monge Monge, Juan Antonio Vargas Núñez, Rosa García Fenoll, Carmen Suárez Fernández, Santiago Jesús Freire Castro, Manuel Mendez Bailon, Isabel Perales Fraile, Manuel Madrazo, Paula Maria Pesqueira Fontan, Jeffrey Oskar Magallanes Gamboa, Andrés González García, Anxela Crestelo Vieitez, Eva María Fonseca Aizpuru, Asier Aranguren Arostegui, Ainara Coduras Erdozain, Carmen Martinez Cilleros, Jose Loureiro Amigo, Francisco Epelde, Carlos Lumbreras Bermejo, Juan Miguel Antón Santos

**Affiliations:** 1Internal Medicine Department, Infanta Cristina University Hospital, 28981 Parla, Spain; irenemariacalo@gmail.com; 2Pathology Department, Infanta Cristina University Hospital, 28981 Parla, Spain; esierrag3@yahoo.es; 3Internal Medicine Department, La Paz University Hospital, 28046 Madrid, Spain; farnalich@salud.madrid.org; 4Internal Medicine Department, Albacete University Hospital Complex, 02008 Albacete, Spain; jlbeato@sescam.org; 5Internal Medicine Department, Segovia Hospital Complex, 40002 Segovia, Spain; dmonge5@hotmail.com; 6Internal Medicine Department, Puerta de Hierro University Hospital, 28222 Majadahonda, Spain; juanantonio.vargas@uam.es; 7Internal Medicine Department, Miguel Servet Hospital, 50009 Zaragoza, Spain; rosa.gfenoll@gmail.com; 8Internal Medicine Department, La Princesa University Hospital, 28006 Madrid, Spain; csuarezfe@gmail.com; 9Internal Medicine Department, A Coruña University Hospital, 15006 A Coruña, Spain; santiago.freire.castro@sergas.es; 10Internal Medicine Department, Clinico San Carlos Hospital, 28040 Madrid, Spain; manuelmenba@hotmail.com; 11Internal Medicine Department, Infanta Sofía Hospital, 28703 San Sebastián de los Reyes, Spain; isabel.perales@salud.madrid.org; 12Internal Medicine Department, Dr. Peset University Hospital, 46017 Valencia, Spain; manel.madrazo@gmail.com; 13Internal Medicine Department, Santiago Clinical Hospital, 15706 Santiago de Compostela, Spain; paulapesqueira@hotmail.com; 14Internal Medicine Department, Nuestra Señora del Prado Hospital, 45600 Talavera de la Reina, Spain; dr990112@hotmail.com; 15Internal Medicine Department, Ramón y Cajal University Hospital, 28034 Madrid, Spain; andres_gonzalez_garcia@hotmail.com; 16Internal Medicine Department, Royo Villanova Hospital, 50015 Zaragoza, Spain; anxela90@gmail.com; 17Internal Medicine Department, Cabueñes Hospital, 33394 Gijón, Spain; evamfonseca@yahoo.es; 18Internal Medicine Department, Urduliz Alfredo Espinosa Hospital, 48610 Urdúliz, Spain; arrobahiru@gmail.com; 19Internal Medicine Department, Santa Marina Hospital, 48004 Bilbao, Spain; ainara.coduraserdozain@osakidetza.eus; 20Internal Medicine Department, HLA Moncloa Hospital, 28008 Madrid, Spain; cmcilleros@hotmail.com; 21Internal Medicine Department, Moisès Broggi Hospital, 08970 Sant Joan Despí, Spain; ahores@gmail.com; 22Internal Medicine Department, Parc Tauli Hospital, 08208 Sabadell, Spain; fepelde@gmail.com; 23Internal Medicine Department, 12 de Octubre University Hospital, 28041 Madrid, Spain; clumbrerasb@gmail.com

**Keywords:** 2019-nCoV, SARS-CoV-2, coronavirus, COVID-19, eosinophil, prognosis

## Abstract

Objectives: A decrease in blood cell counts, especially lymphocytes and eosinophils, has been described in patients with serious Severe Acute Respiratory Syndrome Coronavirus 2 (SARS-CoV-2), but there is no knowledge of their potential role of the recovery in these patients’ prognosis. This article aims to analyse the effect of blood cell depletion and blood cell recovery on mortality due to COVID-19. Design: This work was a retrospective, multicentre cohort study of 9644 hospitalised patients with confirmed COVID-19 from the Spanish Society of Internal Medicine’s SEMI-COVID-19 Registry. Setting: This study examined patients hospitalised in 147 hospitals throughout Spain. Participants: This work analysed 9644 patients (57.12% male) out of a cohort of 12,826 patients ≥18 years of age hospitalised with COVID-19 in Spain included in the SEMI-COVID-19 Registry as of 29 May 2020. Main outcome measures: The main outcome measure of this work is the effect of blood cell depletion and blood cell recovery on mortality due to COVID-19. Univariate analysis was performed to determine possible predictors of death, and then multivariate analysis was carried out to control for potential confounders. Results: An increase in the eosinophil count on the seventh day of hospitalisation was associated with a better prognosis, including lower mortality rates (5.2% vs. 22.6% in non-recoverers, OR 0.234; 95% CI, 0.154 to 0.354) and lower complication rates, especially regarding the development of acute respiratory distress syndrome (8% vs. 20.1%, *p* = 0.000) and ICU admission (5.4% vs. 10.8%, *p* = 0.000). Lymphocyte recovery was found to have no effect on prognosis. Treatment with inhaled or systemic glucocorticoids was not found to be a confounding factor. Conclusion: Eosinophil recovery in patients with COVID-19 who required hospitalisation had an independent prognostic value for all-cause mortality and a milder course.

## 1. Introduction

In December 2019, a pneumonia of unknown origin was described in the city of Wuhan, the capital of Hubei province in China, caused by a novel coronavirus that was later named Severe Acute Respiratory Syndrome Coronavirus 2 (SARS-CoV-2) [1]. The infection was named COVID-19 (coronavirus disease 2019) in February [2], and later labelled a pandemic [3]. As a result of its global spread, overwhelming almost every healthcare system, COVID-19 has become the greatest health emergency of this century. As of 6 December 2020, nearly 66 million COVID-19 cases had been confirmed, and 1,523,656 patients had died. Initially, Europe was one of the most affected continents, with more than 19.986 million cases; Spain accounted for 1,684,647 of those cases [4].

Great effort has been made in describing the clinical and epidemiological features of COVID-19 [5,6,7], however less is known about prognostic factors [8,9,10]. Older male adults and those with diabetes, hypertension, obesity, cardiovascular disease, or chronic respiratory disease are at a greater risk of developing severe COVID-19 [6,8,9,10,11]. Some prognostic factors upon admission are lymphopenia and high levels of D-dimer (DD), lactate dehydrogenase (LDH), and C-reactive protein (CRP) [8,9,12].

Some studies have reported low total eosinophil counts in COVID-19 inpatients and persistently low eosinophil counts in more severe cases [12,13,14,15,16,17,18,19,20]; therefore, eosinopenia upon admission has been proposed as a reliable early diagnostic marker for COVID-19 infection [12,13,14]. A correlation between eosinophil recovery and radiographic and virologic recovery [15,16,17,18] as well as clinical improvement [11,12,13,15] has been more sparingly described; additionally, a worse prognosis when eosinophil levels do not recover has been suggested [11,13,15,19], whereas other studies discarded eosinopenia as a prognostic marker [12]. A meta-analysis of those reports, however, found no effect of eosinophil counts upon admission or eosinophil recovery during the course of COVID-19 [20].

The Spanish Society of Internal Medicine (SEMI, for its initials in Spanish) has launched the SEMI-COVID-19 Network, a collaborative nationwide effort to compile information on patients hospitalised with COVID-19. In a preliminary study of potential prognostic factors (not yet published), recovery from both lymphopenia and eosinopenia correlated with a lower risk of death on a multivariate analysis.

We decided to conduct a specific analysis to demonstrate whether eosinopenia or eosinophil recovery could be a prognostic factor against death due to COVID-19.

### Hypothesis and Objectives

According to our preliminary data, we hypothesised that recovery from eosinopenia could serve as an independent predictor of a favourable outcome in patients with COVID-19.

The primary aim of the study was to evaluate whether eosinophil recovery was a predictive factor of favourable progress during hospitalisation in COVID-19 patients. The secondary aims were: (a) to explore the relationship between recovery from eosinopenia and the development of acute respiratory distress syndrome (ARDS); (b) to evaluate the possible confounding effects of the use of corticosteroids in these patients; and (c) to evaluate the possible confounding effects of prior comorbidities that affect eosinophil counts.

## 2. Methods

### 2.1. Registry Design and Data Collection

The SEMI-COVID-19 Registry is an ongoing, nationwide, retrospective cohort that includes consecutive patients with a confirmed COVID-19 infection who have been hospitalised and discharged from Spanish hospitals. The registry’s characteristics have been thoroughly described in other works [21]. The full list of hospitals and collaborators is shown in Appendix A.

Inclusion criteria for the registry are age ≥18 years and first hospital discharge with a confirmed diagnosis of COVID-19. Exclusion criteria are subsequent admissions of the same patient and denial or withdrawal of informed consent. From 24 March to 29 May 2020, a total of 12,826 discharged patients were included in the registry.

Patients are treated at their attending physician’s discretion, according to local protocols and clinical judgement. Patients included in open-label clinical trials are eligible for inclusion in the registry provided that all information about treatment is available.

Data from medical records are collected retrospectively at discharge by clinical investigators all over the country, using a standardised online data capture system (DCS) described elsewhere [21]. The data collected includes many variables, collected and defined in more detail in the SEMI-COVID-19 Registry [21].

The Spanish Society of Internal Medicine is the sponsor of this registry. The researchers who coordinate the study at each hospital are SEMI members and were asked to participate in this study on a voluntary basis; they did not receive any remuneration for their participation.

The processing of personal data strictly complied with Spanish Law 14/2007, of 3 July, on Biomedical Research; Regulation (EU) 2016/679 of the European Parliament, and of the Council of 27 April 2016, on the protection of natural persons with regard to the processing of personal data and on the free movement of such data, and repealing Directive 95/46/EC (General Data Protection Regulation); and Spanish Organic Law 3/2018, of December 5, on the Protection of Personal Data and the Guarantee of Digital Rights. The SEMI-COVID-19 Registry was approved by the Provincial Research Ethics Committee of Málaga (Spain) on 27 March 2020 (Ethics Committee code: SEMI-COVID-19 27/03/20), and endorsed by the ethics committee of each participant hospital. All patients gave their informed consent. When there were biosafety concerns and/or when the patient had already been discharged, verbal informed consent was requested and noted on the medical record.

### 2.2. Study Design

A retrospective cohort study was designed in order to control for potential confounding variables. Patients included in the SEMI-COVID-19 Registry as of 31 May 2020, were selected for inclusion in this study if they had: (a) all epidemiological data recorded; (b) data on lymphocyte and eosinophil counts upon admission and on the secondary analysis at seven days after admission; and (c) onset of symptoms prior to admission. This last criterion was necessary given that the registry included nosocomial infections and, because laboratory analyses were performed upon admission and on the seventh day of hospitalisation, this ensured that the values did not correlate to clinical progress in nosocomial infections.

Descriptive analysis of the cohort and a multivariate analysis for prognostic factors was performed. Variables that have previously been demonstrated in a literature search to be correlated with eosinophil count (such as asthma or chronic corticoid use) or with COVID-19 severity or progress were considered for multivariate analysis. Variables selected for analysis included demographic variables (age, sex, race, obesity, hypertension, diabetes, alcohol abuse, tobacco use, chronic kidney disease, chronic respiratory diseases, comorbidity burden, degree of dependency, and use of inhaled or systemic corticosteroids); clinical variables (signs and symptoms upon admission, laboratory results and radiographic findings upon admission); treatment received prior to the second laboratory analysis; results of the second laboratory analysis; and clinical outcomes (specifically, pneumonia, ARDS, acute kidney injury, sepsis, ICU admission, and death).

Eosinopenia was defined as a total eosinophil count <150 × 10^6^/L upon admission. Eosinophil recovery was defined as an elevation greater than 80 × 10^6^/L on the second analysis performed on the seventh day of hospitalisation. Lymphopenia was classified into four categories: <800, 800–999, 1000–1199, and ≥1200. Lymphocyte recovery was defined as an elevation greater than 200 × 10^6^/L on the second analysis. Quick sequential organ failure assessment index (qSOFA) values were calculated from the physical findings upon admission.

All other quantitative variables were categorised as normal or abnormal (according to reference levels) upon admission. The evolution of significant values during the hospital stay were categorised as absolute elevation (for D-dimer or glycaemia) or relative elevation (for lactate dehydrogenase (LDH), aspartate aminotransferase (AST), alanine aminotransferase (ALT), and creatinine).

The STROBE Statement guidelines were followed in the conduct and reporting of the study [22].

### 2.3. Statistical Analysis

In the descriptive analysis, we summarised the epidemiological data, demographics and comorbidities, signs and symptoms upon admission, laboratory upon admission, and on the seventh day of hospitalisation, chest radiography findings, treatment received, and clinical outcomes. We performed an initial univariate analysis to determine any differences between eosinophil-recoverers and non-recoverers. We then performed a second univariate analysis to determine factors that correlated with death.

Continuous variables are expressed as means and standard deviation (SD); categorical variables are expressed as absolute values and percentages. We conducted the analysis by means of the Student’s *t*-test or ANOVA test for quantitative variables, and the Chi-squared test or Fisher’s exact test to compare differences between groups. A univariate analysis was performed to explore possible risk factors for death using binomial logistic regression.

Variables associated either with eosinophil recovery (potential confounding factors) or with death were included in a backward-stepwise multivariate logistic regression model for mortality. Survival analysis was deemed unnecessary, because there were no censored cases (each patient was discharged and the date of discharge or death was recorded in the registry), as per the design of the registry design, and time until death or discharge was not considered relevant. Quantitative variables were categorised as normal or abnormal upon admission, and significantly elevated or not significantly elevated at seven days of hospitalisation.

A secondary multivariate analysis was conducted with the composite endpoint of in-hospital death, ICU admission, or onset of moderate-to-severe ARDS.

We used SPSS (IBM Corp. Released 2017. IBM SPSS Statistics for Windows, Version 25.0. Armonk, NY: IBM Corp.) for all analyses.

## 3. Results

### 3.1. Sample Characteristics

The SEMI-COVID-19 Registry included 12,826 patients as of 29 May 2020. Of them, 533 did not have all demographic and epidemiological data recorded (sex, age, race, and date of onset of symptoms), and thus were excluded. Another 510 patients were excluded because their discharge date was not recorded. Of the 11,783 discharged patients with all epidemiological data available, 282 were excluded because they did not have eosinophil counts upon admission, and a further 1455 were excluded for not having eosinophil counts on the seventh day of hospitalisation. Finally, 402 patients had been admitted prior to onset of symptoms and were thus also excluded. A total of 9644 patients fulfilled all inclusion criteria for this study. Of these, 3335 patients (34.6%) had eosinophil recovery, whereas 6309 patients (65.4%) did not. Figure 1 shows the flowchart for patient inclusion.

Demographic and clinical features of the study cohort are described in Table 1**.** There were differences upon admission between patients who showed eosinophil recovery and those who did not. Some important features, such as sex, obesity, or asthma, did not differ. Non-recoverers had a higher overall age and higher rates of hypertension, diabetes, chronic kidney disease (CKD), and chronic obstructive pulmonary disease (COPD). Recoverers, on the other hand, had higher comorbidity burdens and a greater degree of dependency.

Clinical presentation also differed between recoverers and non-recoverers. Recoverers had a longer duration of symptoms prior to admission, higher rates of cough and arthromyalgia, and lower rates of dyspnoea. Confusion and tachypnoea were more frequent in non-recoverers. There were no differences in temperature, heart rate, or arterial systolic tension, but oxygen saturation rates were lower in non-recoverers. A higher proportion of non-recoverers also had a qSOFA index value ≥2.

Non-recoverers had worse laboratory analysis profiles upon admission, with higher glucose, creatinine, D-dimer, and LDH levels. Lymphocyte counts were not significantly different, but eosinophil counts were lower among recoverers. Pulmonary infiltrates on radiological tests were more frequent in eosinophil recoverers.

Treatments and outcomes are summarised in Table 2. Eosinophil-recoverers were more frequently treated with hydroxychloroquine and less frequently treated with systemic or inhaled glucocorticoids. There were no differences between recoverers and non-recoverers regarding treatment with lopinavir–ritonavir, azithromycin, or low-molecular-weight heparin. All outcomes were better among eosinophil-recoverers, with lesser rates of pneumonia, ARDS, acute kidney injury, sepsis, ICU admission, and death. Notably, 94.8% of eosinophil-recoverers were discharged alive (vs 77.4% in non-recoverers, *p* < 0.001), 91.3% were discharged without requiring ICU admission (vs 71.1% in non-recoverers, *p* < 0.001), and 85.8% were discharged with neither ICU admission nor onset of ARDS during hospitalisation (vs 65.5% in non-recoverers, *p* < 0.001).

### 3.2. Outcomes

Variables that correlated with either mortality or eosinophil recovery upon univariate analysis (Table 3), as well as potential confounding factors, were introduced into a multivariate analysis using mortality as the dependent variable. Several cut-off points for categorisation were checked for sensitivity analysis. The final regression model is summarised in Table 4, and shows that eosinophil recovery was independently associated with lower mortality, with an OR of 0.234 (95% CI, 0.154 to 0.354). Initial eosinopenia was not found to be significant in the analysis. A lymphocyte count lower than 800 × 10^6^/L upon admission was predictive of death, but neither further categorisation of lymphocyte value ranges nor lymphocyte recovery were. Corticosteroid treatment was not found to correlate with death in our analysis, whereas both hydroxychloroquine and azithromycin correlated with a lower mortality rate. Notably, both elevated ALT upon admission and at seven days of hospitalisation correlated with a lower mortality rate. More studies are needed to clarify this finding.

A secondary multivariate analysis was performed for the secondary composite endpoint of in-hospital death, ICU admission, or onset of ARDS during hospitalisation (Table 5). After controlling for other variables, eosinophil recovery was found to correlate with a lower chance of worse progress (OR 0.474; 95% CI, 0.383–0.586).

## 4. Discussion

Our study shows that eosinophil recovery has a positive prognostic impact in COVID-19 that is independent of previous lymphocyte or eosinophil levels and previous use of systemic or inhaled corticosteroids. To the best of our knowledge, this work is the first instance where this prognostic factor has been thoroughly described in a large cohort.

Abnormal laboratory values in patients with COVID-19, in particular low levels of lymphocytes, have been described in several studies, but less emphasis has been placed on low levels of eosinophils [8,9,12,16,19]. Lymphocyte depletion has been shown to have diagnostic value, along with prognostic value shown in various studies, albeit inconsistently. The recovery of lymphocytes and eosinophils has been studied to a lesser degree than the implications of their initial values [19]. The first descriptions of eosinophil depletion came from small series [16,18], and eosinopenia upon admission was proposed as a reliable early diagnostic marker for SARS-COV-2 infection [12,13,14]. Eosinopenia has already been described by Echevarria et al. [23] as an independent predictor of death in non-COPD patients with pneumonia, regardless of corticosteroid use. In COVID-19 pneumonia, eosinophil recovery has also been proposed to be a marker for clinical improvement, and sustained eosinopenia a marker of poorer prognosis [11,13,15]. Du Yu also found a correlation with higher viral loads [11]. However, in a series of 414 patients, eosinopenia was not related to prognosis [12], and a previous meta-analysis of 294 subjects [20] had showed that eosinophil levels made no difference in the progress and mortality of patients with COVID-19.

The recovery of lymphocytes and eosinophils has been studied to a lesser degree than the implications of their initial values [14].

In our cohort, which comprised 9644 patients, a profound degree of eosinopenia was found upon diagnosis of COVID-19, with a higher mortality rate observed in patients with eosinopenia patients than patients without eosinopenia (16.7% vs. 13.2%, *p* = 0.04). Furthermore, eosinophil recovery was associated with higher survival rates, as was found by Sun et al. [19]. However, these findings could have been due to a number of confounding factors, the most obvious being that comorbidities or immunosuppressive drugs (used predominantly in more severe cases) could have been responsible for the prolonged eosinopenia, and thus eosinophil recovery would be a marker of other previous prognostic factors. Another explanation could be that eosinophil levels and eosinophil recovery are parallel to lymphocyte levels, representing the same degree of immune response to SARS-CoV-2. The most obvious potential confounding factor is prior use of glucocorticoids, which have been widely described as a cause of eosinopenia by means of medullary retention [12]. For this reason, we designed our study to control for the use of systemic or inhaled glucocorticoids both before and during hospitalisation as a potential confounding factor in sustained eosinopenia and COVID-19 progress.

The multivariate analysis showed no effects of chronic or acute use of corticosteroids, asthma, or other diseases that affect eosinophil levels on the predictive capacity of plasmatic eosinophils. In our analysis, asthma or pulmonary infiltrates on radiological tests did not significantly correlate with mortality and were eliminated from the model. The elevation of eosinophils was found to be associated with a better prognosis and lower mortality rate, with an OR of 0.234 (95% CI, 0.154 to 0.354), independently of previous use of glucocorticoids. Our results are in contrast to the conclusions of the meta-analysis by Lippi et al. [20] or the series of Mamta Soni [12] and corroborate the results of other works [11,13,14,15] where the recovery of eosinopenia is proposed as a potential prognostic factor in COVID-19. All these findings emphasise the incompletely explored role of eosinophils, either as an immunological side effect of the SARS-COV2 virus or as an immunomodulatory factor in COVID-19. Our results could be explained by either distinct initial inflammatory responses to SARS-CoV-2, with an initial predisposition towards a Th2 response, or by different inflammatory evolutions, with an immune recovery with modification from an initial Th1 inflammatory response to a Th2 response [24], or indeed both of them simultaneously.

It has been proposed that SARS-COV2 infections induces eosinopenia. Proposed mechanisms for SARS-COV-2-induced eosinopenia could involve a diminished release of eosinophils from the bone marrow, a blockade in eosinophilopoiesis, direct eosinophil apoptosis induced by dysfunctional type I IFNs response during virus infection, or all of them combined [17]. Eosinophil recovery could thus simply be a marker of lesser virological activity, but this is not probable because it has been found to precede the negativization of nucleic acid assays by five days [15].

Eosinophil recovery could be a marker of a different inflammatory pathways associated with mortality. Several studies have demonstrated the key role of eosinophils in the initiation and maintenance of inflammation through stimulation of a Th2 inflammatory response, as well as their direct association with inflammatory diseases such as asthma [25,26,27,28,29]. Curiously, asthma, which was initially suspected to be a risk factor in COVID-19, has been consistently shown to have a protective role in various cohorts [4,9,17], except for severe asthma, which may be neutrophilic asthma not mediated by a Th2 response. If an underlying Th2 response is involved in eosinophil recovery, it would be expected that we would find a higher proportion of asthmatic patients amongst eosinophil-recoverers and higher levels of eosinophils upon admission. However, in our series, eosinopenia was more severe in eosinophil-recoverers and thus does not suggest a Th2 response prior to admission.

On the other hand, patients with obesity and type 2 diabetes mellitus are known to have a higher Th1 inflammatory response [30,31] and to have a worse COVID-19 prognosis [6,8,9,10,11,32]. Both greater eosinopenia and lower recovery of eosinophil counts could simply be markers of these previous comorbidities; both diabetes and obesity were more prevalent among non-recoverers. Our univariate analysis confirmed higher mortality rates amongst patients with obesity and diabetes, but this effect disappeared in the multivariate analysis. Therefore, it could well be the other way round: instead of eosinophil recovery being a surrogate for lower diabetes rates, the latter could be a deleterious factor because it implies an intrinsic Th1 response, leading to a worse prognosis for COVID-19.

Another possible immunological explanation for the role of eosinophils could be that, regardless of the initial response to SARS-CoV-2 infection, eosinophil recovery represents a marker of immune recovery. This could also be due to a non-specific pathway or to Th2 switching. Were it due to non-specific recovery, it would merely be a marker of good progress with no special immunological significance and should be paralleled or followed by lymphocyte recovery. Our study shows that lymphocyte recovery at the seventh day of hospitalisation is not an independent marker of a good prognosis, whereas Sun et al. [19] found an elevation in lymphocyte counts in less severe cases, albeit starting later than eosinophil recovery. Our database only includes two laboratory analyses (upon admission and on the seventh day); therefore, it was not possible for us to ascertain whether a later lymphocyte recovery exists or if it has prognostic implications. Regardless, a marker of a good prognosis after the seventh day of hospitalisation is probably less useful than an earlier predictor would be.

On the other hand, eosinophil recovery could be a marker of Th2 switching [33,34,35], thus possibly indicating a different inflammatory response to SARS-CoV-2 and leading to less susceptibility to ARDS. This is a highly interesting explanation that should be studied further, because it could well lead to new therapeutic strategies for COVID-19.

Different immunological profiles [33,34,35] have been described in other inflammatory diseases of both autoimmune and infectious origin. The ones most commonly described are the Th1 pathways (involving the so-called Th1 cytokines of IL-12, IFN, and TNF-α, leading to activation of CD8+ T cells and classically activated macrophages), the Th2 pathways (mediated by IL-4, IL-5, and IL-13, leading to activation of eosinophils, alternatively activated macrophages, and B-lymphocytes), and the Th17 pathways (mediated by IL-1, IL-6, and the inflammasome, leading to IL-17 and IL-22) [34,35,36]. In COVID-19, cytokine elevation has been described as a marker of worse progress (higher ARDS and death rates), with involvement serum levels of both IL-1 and IL-6. These patients probably develop a Th1–Th17 response to the infection. A depletion of Treg lymphocytes, which are crucial for the negative regulation of proliferation and inflammation, has been described in COVID-19 patients, especially in more severe cases [37]. There is no knowledge of the mechanism of lung inflammation, because live biopsies have not been described to date. Autopsies after ICU death have shown low-grade inflammation and high rates of local microthrombosis [38], but this may be the advanced, terminal stage of a previous inflammatory injury. Different inflammatory pathways could explain the different progress observed amongst COVID-19 patients. It may not be a question of whether an inflammatory response is provoked, but rather which inflammatory response is provoked. Lessening cytokine dysregulation with immunosuppressants has already been attempted. Perhaps efforts towards inducing a Th2 response could improve patient prognosis but, to our knowledge, there is no pharmacological pathway to do so.

In our study, we explored other changes in the laboratory findings over the course of a patient’s disease. Our multivariate model showed the significance of ALT, LDH, creatinine, and D-dimer elevation. Another finding in our study was the protective effect of both hydroxychloroquine and azithromycin observed in our sample. These findings should be interpreted cautiously so as not to fall in “*Table 2 Fallacy*” [39]; our study was not designed to control for confounding factors of renal or hepatic function.

Finally, we also explored the composite endpoint of in-hospital death, ICU admission, or onset of ARDS during patients’ hospital stays. Eosinophil recovery also correlated favourably with this outcome, with an OR of 0.474 (95% CI, 0.383 to 0.586), meaning that not only was death less frequent among eosinophil-recoverers, but a milder course could be predicted. This is highly important, because if eosinophil recovery is confirmed as a marker of a good prognosis, it could be used to guide decisions regarding discharge in otherwise stable patients. In the context of a pandemic, this could help alleviate the strain on healthcare systems by identifying potential candidates for early discharge.

Among the strengths of the SEMI-COVID-19 Registry and its consequent studies are its multicentre, nationwide design, along with the large number of patients included, which provides strong statistical power for confirming hypotheses. However, for the same reason, all the studies based on the SEMI-COVID-19 Registry have common limitations. Only inpatients were included; therefore, it is not possible to extrapolate our results to outpatients. Information bias could be introduced by either the large number of researchers involved or variability in the availability of data from each hospital. Finally, selection bias could be introduced given the voluntary participation of each centre.

Our study was designed to control for possible confounding factors for abnormal eosinophil values, but some of them could not be controlled due to the nature of the data available in the registry. Transfusion of blood products was not recorded and thus this information was not available for study. The influence of the stress response and hormonal treatment were also not recorded, but should be taken into account when assessing haematological parameters. Bacterial coinfection during, or superinfection after, contracting SARS-CoV-2 could have led to different immune responses. Neither thorough cytokine profiles nor lymphocyte subset panels were obtained, because this registry reflects usual clinical practice and not basic research, therefore inflammatory pathways were not studied. Further research is needed to overcome these limitations.

In conclusion, eosinophil recovery at the seventh day of hospitalisation was a predictor of a good prognosis in COVID-19 inpatients from our cohort, and warrants further research.

## 5. Conclusions

Eosinophil recovery, independently of treatments administered and the patients’ underlying condition, was a marker of good prognosis in our cohort. If confirmed, it could help in making decisions about safe discharge.

More studies are needed to assess whether eosinophil recovery is a marker of general immune recovery or of a different immunological response profile to the infection.

## Figures and Tables

**Figure 1 jcm-10-00305-f001:**
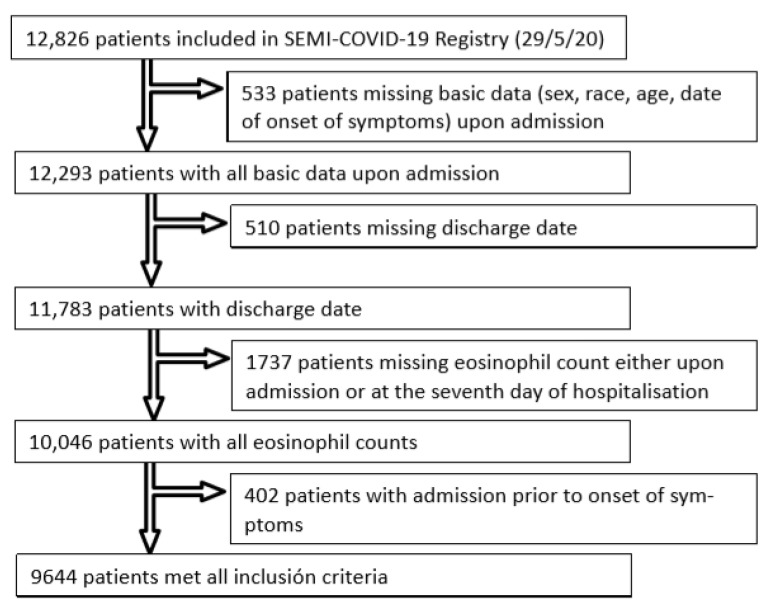
Patient Flowchart.

**Table 1 jcm-10-00305-t001:** Baseline demographic and clinical features upon admission of patients with eosinophil recovery during the course of COVID-19 (Recoverers) and those who did not (Non-Recoverers).

KERRYPNX	Recoverers	Non-Recoverers	*p*
Demographics			
Patients	3335 (34.6)	6309 (65.4)	
Age (years) (*n* = 9644)	63.85 ± 15.57	67.52 ± 16.0	<0.001
Gender (male) (*n* = 5509)	1904 (57.1)	3605 (57.2)	0.479
Race/ethnicity (*n* = 9489)			<0.001
Caucasian	2797 (85.3)	5603 (90.2)	
Latino/a	406 (12.4)	505 (8.1)	
African/Black	15 (0.5)	26 (0.4)	
Alcohol abuse (*n* = 9394)	126 (3.9)	298 (4.8)	0.036
Tobacco use (*n* = 9218)			
Current smoker	156 (4.9)	332 (5.5)	
Former smoker	741 (23.3)	1562 (25.8)	0.009
Degree of dependency (*n* = 9527)			<0.001
Independent or mild	138 (4.2)	439 (7.0)	
Moderate	175 (5.3)	587 (9.4)	
Severe	2986 (90.5)	5202 (83.5)	
Cardiovascular risk factors			
Hypertension (*n* = 9628)	1510 (45.3)	3222 (51.2)	<0.001
Diabetes mellitus (*n* = 9612)	558 (16.8)	1259 (20.0)	<0.001
Obesity (*n* = 8785)	670 (22.0)	1240 (21.6)	0.346
Respiratory diseases			
COPD (*n* = 9619)	159 (4.8)	476 (7.6)	<0.001
Asthma (*n* = 9617)	245 (7.4)	504 (8.0)	0.263
Chronic kidney failure *(n* = 9616)	132 (4.0)	410 (6.5)	<0.001
Comorbidity (*n* = 9366)			<0.001
No comorbidities	373 (11.5)	1158 (18.9)	
Mild	329 (10.2)	850 (13.9)	
Severe	2538 (78.3)	4118 (67.2)	
Previous chronic drug therapy			
Chronic treatment with systemic corticoids (*n* = 9618)	82 (2.5)	313 (5.0)	<0.001
Chronic treatment with inhaled corticoids (*n* = 9578)	262 (7.9)	657 (10.5)	<0.001
Symptoms			
Time from onset of symptoms (days) (*n =* 9644)	7.4 (7.7)	6.7 (4.8)	<0.001
Cough (*n* = 9619)			0.011
No	749 (22.5)	1003 (25.2)	
Dry	2044 (61.4)	3703 (58.9)	
Productive	534 (16.1)	1586 (15.9)	
Dyspnoea (*n* = 9599)	1850 (55.8)	3683 (58.6)	0.007
Arthromyalgia (*n* = 9539)	1172 (35.5)	1910 (30.6)	<0.001
Asthenia (*n* = 9513)	1501 (45.6)	2705 (43.5)	0.048
Anorexia (*n* = 9482)	626 (19.1)	1265 (20.4)	0.123
Fever at home (*n* = 9608)			0.065
<37 °C	448 (13.5)	915 (14.6)	
37.0–37.9 °C	651 (19.6)	1307 (20.8)	
>38.0 °C	2230 (67.0)	4057 (64.6)	
Physical examination at admission			
Confusion (*n* = 9530)	223 (6.8)	740 (11.9)	<0.001
Tachypnoea (>20 brpm) (*n* = 9394)	848 (26.0)	2029 (33.1)	<0.001
SBP (mmHg) (*n* = 9252)	128.5 ± 20.28	128.8 ± 21.3	0.538
Heart rate (bpm) (*n* = 9351)	88.8 ± 16.90	88.6 ± 17.5	0.581
Temperature (°C) (*n* = 9345)	37.1 ± 0.98	37.1 ± 0.98	0.253
Oxygen saturation (*n* = 9410)	93.8 ± 4.45	92.9 ± 5.7	<0.001
Saturation <95%	1550 (47.5)	3197 (52)	<0.001
Lung auscultation			
Crackles (*n* = 9416)	1726 (53.1)	3290 (53.3)	0.854
Wheezing (*n* = 9414)	143 (4.4)	425 (6.9)	<0.001
qSOFA score (*n* = 9644)			<0.001
0–1	3196 (95.8)	5799 (91.9)	
2–3	139 (4.2)	510 (8.1)	
Additional tests			
Radiological findings			
Interstitial pulmonary infiltrates (*n* = 9600)			<0.001
No pulmonary infiltrates	348 (10.5)	853 (13.6)	
Unilateral pulmonary infiltrates	726 (21.9)	1308 (20.8)	
Bilateral pulmonary infiltrates	2247 (67.7)	4118 (65.5)	
Laboratory findings upon admission			
PO2/FiO2 ratio (mmHg) (*n* = 4859)	303.3 ± 94.7	288.5 ± 98.6	<0.001
Leukocytes × 10^6^/L (*n* = 9644)	7262 ± 5002	7192 ± 5562	0.538
Eosinophils × 10^6^/L (*n* = 9644)	18.25 ± 64.13	37.45 ± 107.64	<0.001
Eosinopenia <150 × 10^6^/L	3252 (97.5)	5906 (93.6)	<0.001
Lymphocytes × 10^6^/L *(n* = 9644)	1126 ± 1562	1098 ± 1805	0.443
Lymphopenia <800 × 10^6^/L	939 (28.2)	2312 (36.6)	<0.001
Lymphopenia <800 × 10^6^/L			
Lymphopenia <800 × 10^6^/L			
Neutrophils × 10^6^/L (*n* = 9644)	5230 ± 2927	5192 ± 3382	0.583
CRP (mg/L) (*n* = 9285)	82.2 ± 80	85.2 ± 86.3	0.097
Glucose (mg/dL) (*n* = 9368)	123.7 ± 52.9	127.3 ± 57.7	0.003
Creatinine (mg/dL) (*n* = 9614)	1.0 ± 0.69	1.11 ± 0.86	<0.001
Urea (mg/dL) (*n* = 7713)	41.8 ± 31.5	48.3 ± 36.6	<0.001
LDH (U/L) (*n* = 8448)	341.2 ± 155.1	355.7 ± 179.3	<0.001
AST (U/L) (*n* = 7616)	47.4 ± 48.3	47.7 ± 59.1	0.847
ALT (U/L) (*n* = 9120)	42.2 ± 42.0	41.2 ± 52.4	0.373
D-dimer (ng/mL) (*n* = 7567)	1354.8 ± 5157	1619.7 ± 5548	0.043

COPD: chronic obstructive pulmonary disease. Comorbidity was measured using the Charlson Comorbidity Index. brpm: breaths per minute. SBP: systolic blood pressure. mmHg: millimetres of mercury. bpm: beats per minute. qSOFA: quick sequential organ failure assessment. CRP: C-reactive protein. LDH: lactate dehydrogenase. AST: aspartate aminotransferase. ALT: alanine aminotransferase. Categorical variables are expressed as N (%), quantitative variables as mean ± SD.

**Table 2 jcm-10-00305-t002:** Management and progress during hospitalisation of patients with and without eosinophil elevation.

	Recoverers	Non-Recoverers	*p*
Treatment Received			
LPV/r (*n* = 9606)	2151 (64.6)	4047 (64.5)	0.868
Hydroxychloroquine (*n* = 9617)	3016 (90.5)	5432 (86.4)	<0.001
Systemic corticosteroids (*n* = 9644)	458 (13.7)	2003 (31.7)	<0.001
Tocilizumab (*n* = 9578)	274 (8.2)	664 (10.6)	<0.001
Azithromycin (*n* = 9592)	2079 (62.6)	3911 (62.4)	0.501
Inhaled corticosteroids (*n* = 9497)	164 (5.0)	400 (6.5)	0.004
LMWH (*n* = 9564)	2804 (84.5)	5256 (84.1)	0.611
Outcomes			
Pneumonia (*n* = 9605)	257 (7.7)	791 (12.6)	<0.001
ARDS (*n* = 9595)			<0.001
No	2605 (78.3)	3933 (62.7)	
Mild	299 (9.0)	547 (8.7)	
Moderate	156 (4.7)	531 (8.5)	
Severe	266 (8.0)	1258 (20.1)	
Acute kidney failure (*n* = 9613)	315 (9.5)	985 (15.7)	<0.001
Sepsis (*n* = 9604)	102 (3.1)	462 (7.4)	<0.001
ICU admission (*n* = 9636)	179 (5.4)	678 (10.8)	<0.001
Length of hospital stay (days) (*n* = 9644)	11.0 ± 7.8 (*)	11.5 ± 9.2 (*)	<0.001
Death (in-hospital) (*n* = 9644)	172 (5.2)	1423 (22.6)	<0.001
Composite endpoint (in-hospital death or ICU admission or ARDS) (*n* = 9612)	472 (14.2)	2170 (34.5)	<0.001
Discharge	3163 (94.8)	4886 (77.4)	<0.001
without ICU admission	3040 (91.3)	4484 (71.1)	
without ICU admission or ARDS	2852 (85.8)	4118 (65.5)	

LPV/r: lopinavir/ritonavir. LMWH: low-molecular-weight heparin. ARDS: acute respiratory distress syndrome. ICU: intensive care unit. Categorical variables are expressed as N (%), quantitative variables (*) as mean ± SD.

**Table 3 jcm-10-00305-t003:** Univariate analysis of mortality. Quantitative variables are expressed as mean ± SD in survivors and non-survivors. Categorical variables are expressed as mortality in N (%) for factor present and factor absent. For categorical variables with more than two categories, mortality is provided for each category as N (%).

	Non-Survivors	Survivors	*p*
Age (years)	78.7 ± 10.5	63.8 ± 15.7	<0.001
Time from onset of symptoms at admission (days)	5.7 ± 5.0	7.2 ± 6.1	<0.001
Length of hospital stay (days)	10.5 ± 9.4	11.5 ± 8.6	<0.001
Factor	Mortality when present	Mortality when absent	
Demographics			
Male gender	1011 (18.4)	581 (14.1)	<0.001
Caucasian race/ethnicity	1503 (17.9)	72 (6.6)	<0.001
Alcohol abuse	86 (20.3)	1468 (16.4)	0.034
Tobacco use	585 (21.0)	936 (14.6)	<0.001
Moderate or severe dependency	531 (39.7)	1043 (12.7)	<0.001
Hypertension	1115 (13.6)	477 (9.7)	<0.001
Obesity	359 (18.8)	1067 (15.5)	0.001
Diabetes mellitus	480 (26.4)	1111 (14.3)	<0.001
COPD	200 (31.5)	1390 (15.5)	<0.001
Asthma	89 (11.9)	1500 (16.9)	<0.001
Chronic kidney disease	200 (36.9)	1389 (15.3)	<0.001
Moderate or severe comorbidity	523 (34.2)	1023 (13.1)	<0.001
Chronic treatment with systemic corticosteroids	113 (28.6)	1476 (16.0)	<0.001
Chronic treatment with inhaled corticosteroids	208 (22.6)	1372 (15.8)	<0.001
Symptoms			
Cough	1093 (15.0)	494 (21.2)	<0.001
Dyspnoea	1100 (19.9)	485 (11.9)	<0.001
Arthromyalgia	309 (10.0)	1261 (19.5)	<0.001
Asthenia	640 (15.2)	924 (17.4)	0.004
Anorexia	373 (19.7)	1182 (15.6)	<0.001
Fever at home	1266 (15.4)	315 (23.1)	<0.001
Physical examination			
Confusion	439 (45.6)	1135 (13.2)	<0.001
Tachypnoea >20 brpm	848 (29.5)	699 (10.7)	<0.001
Hypotension (<90 mmHg)	56 (36.4)	1492 (16.4)	<0.001
Tachycardia >100 bpm	335 (16.0)	1215 (16.8)	0.383
Temperature >37.7 °C	431 (17.5)	1106 (16.1)	0.115
Oxygen saturation via pulse oximetry (%)			<0.001
Normal (>94%)	375 (8.0)		
Hypoxemia (90–94%)	551 (16.9)		
Desaturation (<90%)	628 (42.2)		
Crackles	943 (18.8)	603 (13.7)	<0.001
Wheezing	152 (26.8)	1392 (15.7)	<0.001
qSOFA score ≥2	333 (51.3)	1262 (14.0)	<0.001
Findings upon admission			
Pulmonary infiltrates on radiological tests	1417 (16.9)	173 (14.4)	0.320
Eosinophils (×10^6^/L)			0.084
>300	26 (15.1)		
150–299	38 (12.1)		
<150	1531 (16.7)		
Eosinopenia <150 × 10^6^/L	1531 (16.7)	64 (13.2)	0.040
Lymphocytes (×10^6^/L)			<0.001
>1200	337 (10.9)		
1000–1199	184 (12.1)		
800–999	269 (15.0)		
<800	805 (24.8)		
Lymphopenia <800 × 10^6^/L	805 (24.8)	790 (12.4)	<0.001
Basal glucose >125 mg/dL	792 (27.4)	761 (11.8)	<0.001
High creatinine (>1.4 mg/dL)	568 (42.2)	1025 (12.4)	<0.001
LDH >360 U/L	702 (23.1)	585 (10.8)	<0.001
AST >60 U/L	322 (21.4)	893 (14.6)	<0.001
ALT >60 U/L	193 (13.0)	1246 (16.3)	0.001
D-dimer (ng/mL)			<0.001
<500	269 (8.9)		
500–999	289 (12.4)		
>1000	535 (24.1)		
D-dimer >1000 ng/mL	535 (24.1)	558 (10.4)	<0.001
Treatment			
Lopinavir/ritonavir	929 (15.0)	656 (19.2)	<0.001
Hydroxychloroquine	1246 (14.7)	341 (29.2)	<0.001
Systemic corticosteroids	532 (21.6)	1063 (14.8)	<0.001
Tocilizumab	214 (22.8)	1372 (15.9)	<0.001
Azithromycin	914 (15.3)	664 (18.4)	<0.001
Inhaled corticosteroids	117 (20.7)	1446 (16.2)	0.005
Low-molecular-weight heparin	1324 (16.4)	252 (16.8)	0.753
Findings during progress			
Eosinophils increased >80 × 10^6^/L	172 (5.2)	1423 (22.6)	<0.001
Lymphocyte increased >200 × 10^6^/L	288 (6.2)	1307 (26.1)	<0.001
LDH increased >50%	349 (48.9)	760 (11.0)	<0.001
Creatinine increased >50%	224 (62.0)	1350 (14.7)	<0.001
D-dimer increased >500 ng/mL	339 (26.8)	491 (9.8)	<0.001
Glycaemia increased >100 mg/dL	136 (42.6)	1353 (15.6)	<0.001
AST increased 3×	72 (24.4)	1045 (15.5)	<0.001
ALT increased 3×	125 (13.4)	1228 (15.9)	0.040

COPD: chronic obstructive pulmonary disease. Comorbidity was measured using the Charlson Comorbidity Index. brpm: breaths per minute. mmHg: millimetres of mercury. bpm: beats per minute. qSOFA: quick sequential organ failure assessment. CRP: C-reactive protein. LDH: lactate dehydrogenase. AST: aspartate aminotransferase. ALT: alanine aminotransferase. Quantitative variables are expressed as mean ± SD in survivors and non-survivors. Categorical variables are expressed as mortality in N (%) for factor present and factor absent. For categorical variables with more than two categories, mortality is provided for each category as N (%).

**Table 4 jcm-10-00305-t004:** Multivariate analysis of mortality. The effect of each factor is expressed as an adjusted odds ratio (CI 95%).

	Adjusted OR	*p*
Demographics		
Age (years)	1.050 (1.036 to 1.065)	0.000
Gender (female)	0.644 (0.471 to 0.881)	0.006
Hypertension	1.320 (0.996 to 1.816)	0.087
Moderate-to-severe dependency	2.250 (1.515 to 3.342)	0.000
Clinical manifestations at admission		
Cough	0.670 (0.483 to 0.929)	0.016
Confusion	1.718 (1.149 to 2.569)	0.008
Tachypnoea	1.894 (1.397 to 2.566)	0.000
Wheezing	1.597 (0.966 to 2.639)	0.068
Desaturation		
Saturation 90–94%	1.701 (1.196 to 2.420)	0.003
Saturation <90%	4.594 (3.084 to 6.843)	0.000
Treatment during hospitalisation		
Hydroxychloroquine	0.662 (0.432 to 1.013)	0.057
Azithromycin	0.647 (0.475 to 0.881)	0.006
Laboratory findings at admission		
Creatinine >1.4 at admission	1.564 (1.103 to 2.219)	0.012
LDH >360 at admission	2.450 (1.757 to 3.416)	0.000
AST >60 at admission	2.462 (1.637 to 3.704)	0.000
ALT >60 at admission	0.444 (0.274 to 0.720)	0.001
Glycaemia >125 at admission	1.405 (1.045 to 1.889)	0.024
Lymphopenia <800 × 10^6^/L at admission	1.452 (1.086 to 1.942)	0.012
Laboratory findings on the seventh day of hospitalisation		
Eosinophil counts increased >80 × 10^6^/L	0.234 (0.154 to 0.354)	0.000
LDH increased >1.5×	10.614 (7.101 to 15.867)	0.000
Creatinine increased >1.5×	6.032 (3.528 to 10.315)	0.000
D-dimer increased >500	2.341 (1.718 to 3.189)	0.000
ALT increased >3×	0.536 (0.321 to 0.894)	0.017

**Table 5 jcm-10-00305-t005:** Multivariate analysis of the composite endpoint of in-hospital death or ICU admission or moderate-to-severe ARDS. The effect of each factor is expressed as an adjusted odds ratio (CI 95%).

	Adjusted OR	*p*
Demographics		
Race (Caucasian)	0.715 (0.528 to 0.969)	0.030
Clinical manifestations at admission		
Duration of symptoms at admission (days)	0.962 (0.942 to 0.983)	0.000
Cough	1.070 (0.856 to 1.337)	0.055
Confusion	1.783 (1.320 to 2.409)	0.000
Tachypnoea >20 brpm	2.057 (1.697 to 2.495)	0.000
Wheezing	1.402 (0.987 to 1.991)	0.059
Fever	1.375 (1.125 to 1.681)	0.002
Desaturation		
Saturation 90–94%	1.694 (1.377 to 2.084)	0.000
Saturation <90%	4.856 (3.730 to 6.322)	0.000
Treatment during hospitalization		
Hydroxychloroquine	0.684 (0.504 to 0.928)	0.015
Corticosteroids	1.634 (1.348 to 1.979)	0.000
Laboratory findings at admission		
Creatinine >1.4 at admission	1.497 (1.162 to 1.928)	0.002
D-dimer >1000 at admission	1.226 (1.006 to 1.495)	0.044
LDH >360 at admission	2.306 (1.907 to 2.790)	0.000
Glycaemia >125 at admission	1.386 (1.143 to 1.681)	0.001
Lymphopenia <800 × 10^6^/L at admission	1.541 (1.222 to 1.944)	0.000
Any pulmonary infiltrates	2.306 (1.601 to 3.321)	0.000
Laboratory findings on the seventh day of hospitalisation		
Eosinophil counts increased >80 × 10^6^/L	0.474 (0.383 to 0.586)	0.000
LDH increased >1.5×	6.437 (4.779 to 8.669)	0.000
Creatinine increased >1.5×	3.485 (2.160 to 5.620)	0.000
D-dimer increased >500	2.643 (2.155 to 3.241)	0.000
Glycaemia increased >100 mg/dL	1.661 (1.083 to 2.548)	0.020

## Data Availability

Data is contained within the article.

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
