# Peer review of "The Prognostic Value of Eosinophil Recovery in COVID-19: A Multicentre, Retrospective Cohort Study on Patients Hospitalised in Spanish Hospitals"

_jcm, 2021, doi:10.3390/jcm10020305_

Round 1

Reviewer 1 Report

This is a very interesting article and a new approach.The strengths are the large population of patients across multiple centres and the detailed analysis of the data. Tracking the eosinophil count is also readily available to bedside physicians and it can easily be adopted into routine care. It is also particularly pertinent to more resource poor settings to assess the trajectory of patient's recovery. 

Author Response

Response to Reviewer 1 Comments

Dear Reviewer,

Thank you very much for your work on our manuscript. Your review has been very useful.

Best regards,

Dra. Mateos and Dr. Antón

Reviewer 2 Report

This study explores, using a large registry dataset, the relationship between eosinophil count recovery and clinical outcomes in a large cohort of COVID-19 patients in Spain.  The authors identified a strong association between eosinophil count recovery and a favourable outcome (e.g. mortality).  The study strengths include the use of a large sample size and a detailed analysis which includes multiple confounding variables.  However, the authors have overstated the significance of their findings – they claim that eosinophil count recovery is predictive of good outcomes and therefore can be used to guide clinical management (e.g. discharge decision). This interpretation is not supported by their findings, based on a major methodological limitation identified in this manuscript, as explained below.

To claim that a parameter is predictive of a predefined outcome, the standard practice in the field usually requires a step-wise process consisting of two independent stages. The first stage involves identifying a statistical association between a variable (e.g. eosinophil recovery) and a predefined outcome (e.g. mortality). This is done using a retrospective patient cohort. The second stage is a validation step, which requires the use of an independent, external dataset to replicate the initial findings. This is usually done using a prospective patient cohort; this cohort needs to be separated from the original cohort. In some instances, this second stage needs to be repeated across multiple patient cohorts to ensure the predictive performance of the identified variable is reproducible across different patient populations in different clinical settings. This second stage is important because it ensures the findings are robust, consistent and generalisable to other patient groups. In this manuscript, there is no evidence that the authors have performed such a validation.

It is important to recognise that the findings in this manuscript represent an association between eosinophil recovery and mortality.  One cannot extrapolate this association to the claim that eosinophil recovery will predict outcome in a future cohort of COVID-19 patients. The authors have made such an extrapolation repeatedly in both the Abstracts and the main text.  This over-claim should be tuned down to avoid misleading the readers. Notably, the authors did not follow the TRIPOD guidelines, which is an established consensus guideline for conducting a prognostic study.  It is therefore no surprise that the authors are unaware of the methodological requirement for additional validation of their findings.

Reviewer 3 Report

This manuscript by Gonzalez et al concerns the evaluation of eosinophil recovery on COVID-19 outcomes in a large (9644 hospitalized patients), multicenter retrospective cohort in Spanish hospitals (147 total). The authors found that a recovery of eosinophil counts during the course of hospitalized care was associated with better outcomes. This is a retrospective study which gains strength from a large cohort size across a relatively large geographic region. The authors take care to analyze their data carefully, and this report will be of interest to a wide audience. That being said, I have some concerns which the authors are encouraged to address:

1. It is unclear why the manuscript does not analyze the relationship between eosinopenia and death or time until death. The authors simply state that it was “not considered relevant.” However, the authors are arguing that recovery from eosinopenia is a prognostic indicator of recovery and may guide discharge guidelines. It would logically follow that there is prognostic value of clinical decline in lack of recovery from eosinopenia, but that does not necessarily mean it is prognostic for determination of potential mortality without statistical analysis. Can the authors please justify further why this measure was excluded, or provide these analyses either in the manuscript or the supplemental?

2. In cases where multiple categories are lumped into the same variable, it is not clear what statistical significance means (e.g., race/ethnicity, degree of dependency, comorbidity, etc), except where the numbers are so skewed as to be obvious. Why did the authors not provide significance for each individual category of stratification?

3. For previous drug therapy using corticoids, is this limited to those patients for whom corticoids were prescribed specifically for COVID-19? i.e., not patients who have otherwise been prescribed corticoids?

4. The finding of an elevated admissions or progress to elevated ALT as being protective is a curious one, which the authors appropriately point out as needing further studies. However, this reviewer is curious why this measure was dropped from the second multivariate analysis with composite endpoints?

5. There is a notable paucity of references in the discussion of immunological explanations for the study’s findings in the discussion. The authors are strongly encouraged to provide more rigorous cited support for their discussion.

6. Whether by missing in literature search, or by time of preparation, there are numerous relevant studies which the authors have not included in their introduction and/or discussion of this paper. The authors are encouraged to read and contextualize their study with the following:

(a) “Eosinopenia and COVID-19”, Tanni et al JAOA 2020, https://doi.org/10.7556/jaoa.2020.091

(b) “Eosinopenia as an early diagnostic marker of COVID-19 at the time of the epidemic", Zhengyuan Xia EClinical Medicine 2020, https://doi.org/10.1016/j.eclinm.2020.100398

(c) “Clinical Features of 85 Fatal Cases of COVID-19 from Wuhan” Du et al., AJRCCM 2020, https://doi.org/10.1164/rccm.202003-0543OC

(d) “The role of peripheral blood eosinophil counts in COVID‐19 patients” Xie et al., Allergy 2020, https://doi.org/10.1111/all.14465

Minor

1. The authors are encouraged to update the numbers of the COVID-19 pandemic effects in the first paragraph, which is now nearly 6 months old and does not capture the acceleration of cases seen in the last half of 2020.

2. Citation error on line 124.

3. A reference for the STROBE Statement guidelines is missing.

Round 2

Reviewer 2 Report

The authors have adequately addressed my concerns. I have no further comments.